# Bile Acids Promote Hepatic Biotransformation and Excretion of Aflatoxin B1 in Broiler Chickens

**DOI:** 10.3390/toxins15120694

**Published:** 2023-12-09

**Authors:** Liang Chen, Tian Wen, Aizhi Cao, Jianmin Wang, Hua Pan, Ruqian Zhao

**Affiliations:** 1MOE Joint International Research Laboratory of Animal Health and Food Safety, Nanjing Agricultural University, Nanjing 210095, China; chenliang202312@163.com (L.C.); wentianayg@163.com (T.W.); 2Key Laboratory of Animal Physiology & Biochemistry, College of Veterinary Medicine, Nanjing Agricultural University, Nanjing 210095, China; 3Huaihua Institute of Agricultural Sciences, Huaihua 418000, China; 4Industrial Research Institute of Liver Health & Homeostatic Regulation, Shandong Longchang Animal Health Product Co., Ltd., Dezhou 253000, China; aizhicao78@163.com (A.C.);; 5National Key Laboratory of Meat Quality Control and Cultured Meat Development, Nanjing 210095, China

**Keywords:** bile acids, aflatoxin B1, detoxification system, liver X receptor α

## Abstract

Aflatoxin B1 (AFB1) is a hazardous mycotoxin that often contaminates animal feed and may potentially induce severe liver damage if ingested. The liver is the primary organ responsible for AFB1 detoxification through enzyme-catalyzed xenobiotic metabolism and bile acid (BA)-associated excretion. In this study, we sought to investigate whether exogenous BA improves hepatic AFB1 detoxification to alleviate AFB1-induced liver injury in broiler chickens. Five-day-old broiler chicks were randomly assigned to three groups. CON and AFB1 received a basal diet; AFB1 + BA received a basal diet with 250 mg/kg BA for 20 days. After a 3-day pre-feed, AFB1 and AFB1 + BA were daily gavaged with 250 μg/kg BW AFB1, while CON received gavage solvent for AFB1 treatment. Dietary BA supplementation protected chickens from AFB1-induced hepatic inflammation and oxidative stress. The hepatic biotransformation of AFB1 to its metabolite AFBO was improved, with accelerated excretion to the gallbladder and cecum. Accordantly, AFB1-induced down-regulation of detoxification genes, including cytochrome P450 enzymes, glutathione S-transferases, and the bile salt export pump, was rescued by BA supplementation. Moreover, liver X receptor α, suppressed by AFB1, was enhanced in BA-treated broiler chickens. These results indicate that dietary BA supplementation improves hepatic AFB1 detoxification and excretion through LXRα-involved regulation of xenobiotic enzymes.

## 1. Introduction

Aflatoxin B1 (AFB1) is considered the most recurrent and dangerous mycotoxin, primarily produced by the fungi *Aspergillus flavus* and *Aspergillus parasitica* [1,2]. After exposure, AFB1 is primarily taken up by hepatocytes from sinusoidal blood [3,4], causing a range of serious health issues, including hepatotoxicity, genotoxicity, and immunosuppression [5]. AFB1 is a major concern for food safety and public health, as it not only contaminates food crops but also remains as residue in human food derived from various farm animals, including poultry [1,6,7].

AFB1 is an inert compound, and its toxicity is dependent on bioactivation (phase I) and detoxification (phase II) pathways. The liver is the primary detoxification organ for AFB1 [1,2]. AFB1 metabolism in the liver is mainly accomplished by CYP450 enzymes, which are responsible for various reactions, including O-demethylation, ketone reduction, hydroxylation, and epoxidation [8]. AFBO, mainly catalyzed by CYP1A1, CYP1A2, and CYP2A6 in chicken [9,10], is a primary and highly toxic AFB1 metabolite due to its ability to react with DNA and proteins, forming harmful adducts. Importantly, a critical pathway for AFB1 detoxification is the conjugation of AFBO with glutathione (GSH) by glutathione S-transferases (GSTs) [8,11]. The metabolites of AFB1, including AFBO-GSH, are primarily excreted into the intestine through bile flow and eventually eliminated from the body [3].

The expression of detoxification enzymes and xenobiotic transporters is regulated by nuclear factors at the level of transcription [12]. Liver X receptor α (LXRα) is considered a key transcription factor for the transactivation of detoxification enzymes and xenobiotic transporters, as it has been reported that human LXRα bidirectionally transactivates CYP1A1 and CYP1A2 through two common cis-elements located on the overlapping 5′-flanking region of these two genes [13], and LXRα agonists activate transcription of the multidrug resistance-associated protein 2 gene that participates in hepatobiliary excretion of endogenous and xenobiotic compounds [14].

Bile acids (BAs) are a class of structurally diverse molecules with detergent properties. They are synthesized from cholesterol within hepatocytes and subsequently discharged into the duodenum via the biliary system [15]. BAs exert their metabolic regulation, anti-inflammation, and anti-oxidative stress functions through binding to their receptors, including the farnesoid X receptor (FXR) [16], the G protein-coupled receptor (TGR5) [17], or LXRα [18]. BAs are reported to mitigate the toxicity of xenobiotics encompassing bilirubin [19] and endotoxins [20] in mice and T-2 in chickens [21] by upregulating the activities of xenobiotic metabolic enzymes or efflux transporters. This study aimed to reduce aflatoxicosis in broilers, with reported evidence indicating that sea buckthorn oil [22] and hydrated sodium calcium aluminosilicate [23] through reducingreduce the residues of AFB1, stimulate hepatic protein synthesis, or enhance immune function in the liver. However, it remains unknown whether exogenous BAs alleviate AFB1-induced liver injury in broiler chickens and whether the detoxification effect of BA, if any, involves the regulation of LXRα and its target genes responsible for hepatic AFB1 biotransformation and excretion.

Here, we use broiler chickens as a model to investigate whether dietary supplementation of a compound BA preparation could alleviate AFB1-induced inflammation and oxidative stress and whether this is associated with LXRα-related regulation of hepatic detoxification genes, including bioactivation (phase I) and detoxification (phase II) enzymes, as well as ABC transporters responsible for AFB1 excretion.

## 2. Results

### 2.1. Dietary BA Supplementation Alleviates AFB1-Induced Morphological Changes Induced by AFB1 in the Liver and Gallbladder

No significant differences among the three groups were observed in body weight, liver weight, or liver weight relative to body weight (Appendix A). Morphological examination of liver sections revealed more infiltrating immune cells in the AFB1 group, which was significantly reduced in the AFB1 + BA group (Figure 1A,B). Interestingly, the gall bladder weight was significantly (*p* < 0.05) reduced in the AFB1 group, which was significantly (*p* < 0.05) restored in the AFB1 + BA group (Figure 1C,D). Moreover, BA supplementation significantly increased the bile volume as compared with both CON and AFB1 groups (Figure 1E).

### 2.2. Dietary BA Supplementation Alleviates AFB1-Induced Hepatic Inflammation and Oxidative Stress

BA supplementation significantly (*p* < 0.05) protected against AFB1-induced hepatic upregulation (*p* < 0.05) of IL-1β, IFNγ, and iNOS mRNA (Figure 2A). IL-1β concentrations in serum (Figure 2B) and liver (Figure 2C) showed the same pattern, suggesting the anti-inflammatory effects of BA supplementation. Additionally, compared to the CON group, there were no significant differences in T-AOC (Figure 2D) and MDA (Figure 2E), while AFB1 induced a significant reduction (*p* < 0.05) in the activities of CAT (Figure 2F) and SOD (Figure 2G). Furthermore, the AFB1 + BA group exhibited a significant increase (*p* < 0.05) in CAT and SOD activities, with no significant differences in T-AOC and MDA when compared to the AFB1 group.

### 2.3. Dietary BA Supplementation Promotes the Turnover of AFB1/AFBO and Bile Acids

Compared to the CON group, the AFB1 group exhibited a significant increase (*p* < 0.05) in AFB1 levels in serum (Figure 3A) and liver (Figure 3B). In contrast, BA supplementation significantly reduced (*p* < 0.05) AFB1 levels in both serum and liver compared to the AFB1 group. Additionally, in comparison to the AFB1 group, the AFB1 + BA group showed a significant increase (*p* < 0.05) in AFB1 concentrations in bile (Figure 3C) and cecum content (Figure 3D), suggesting that BA supplementation promotes the excretion of AFB1. Interestingly, AFBO displayed a distinct metabolic pattern. Compared to the CON group, AFBO concentrations significantly increased (*p* < 0.05) in serum (Figure 3E) but decreased significantly (*p* < 0.05) in the liver (Figure 3F) in the AFB1 group. On the contrary, compared with the AFB1 group, the AFB1 + BA group exhibited a significant decrease (*p* < 0.05) in serum and an increase (*p* < 0.05) in AFBO concentration in the liver. Furthermore, compared to the AFB1 group, the AFB1 + BA group showed a significant increase (*p* < 0.05) in AFBO concentrations in both bile (Figure 3G) and cecum content (Figure 3H) (*p* < 0.05). These results suggest that dietary BA supplementation modulates the metabolism and excretion of AFB1 in the liver. Additionally, serum TBA concentration (Figure 3I) was not affected by AFB1 or BA. However, TBA concentration was significantly (*p* < 0.05) higher in the liver (Figure 3J) yet significantly lower (*p* < 0.05) in the bile (Figure 3K) in the AFB1 group, which were both significantly reversed (*p* < 0.05) in the AFB1 + BA group. No differences among groups were detected in TBA concentration or cecum content (Figure 3L). Regression analyses between AFB1/AFBO and TBA concentrations revealed significant (*p* < 0.001) negative correlations in serum (Figure 3M), liver (Figure 3N), and bile (Figure 3O), respectively. The cecal concentration of AFB1 was not significantly correlated with TBA concentration, yet a tendency toward positive correlation was observed between AFBO and TBA in cecum content (r^2^ = 0.33, *p* = 0.08, Figure 3P). These results suggest that dietary BA supplementation may promote AFB1 and AFBO excretion in association with hepatobiliary TBA efflux in broiler chickens.

### 2.4. Dietary BA Supplementation Upregulates Hepatic Expression of Detoxification Enzymes

AFB1 challenge significantly (*p* < 0.05) downregulated hepatic mRNA expression of phase I detoxification enzymes such as CYP1A1 (Figure 4A), CYP1A2 (Figure 4B), and CYP3A4 (Figure 4C) in broiler chickens. Such AFB1-induced downregulation was significantly (*p* < 0.05) reversed in the AFB1 + BA group, except for the CYP2A6. Moreover, BA supplementation significantly (*p* < 0.05) increased CYP2A6 (Figure 4D) mRNA levels in the liver. The phase II detoxification enzyme GST was also significantly inhibited in the liver of AFB1-challenged chickens at the level of both mRNAs (Figure 4E) and enzyme activity (Figure 4F). BA supplementation significantly alleviated AFB1-induced repression of GST activity in the liver. Accordantly, hepatic mRNA expression of GSTA2 (Figure 4G) and GSTCD (Figure 4H) was significantly (*p* < 0.05) increased in the AFB1 + BA group. Meanwhile, the levels of BSEP mRNA (Figure 4I) and protein (Figure 4J) were significantly (*p* < 0.05) reduced in the AFB1 group, which was significantly restored (*p* < 0.05) in the AFB1 + BA group. Furthermore, while BA supplementation led to an increase in the mRNA expression of MDR1, no significant differences were observed among the three groups in the mRNA or protein expression levels of MRP2, MRP3, and MRP4 (Appendix A). These results suggest that dietary BA supplementation improves AFB1 metabolism and excretion in broiler chickens via modulating hepatic expression of genes encoding detoxification enzymes.

### 2.5. Dietary BA Supplementation Enhances Hepatic Expression of LXRα

Among five transcription factors reported to modulate hepatic detoxification genes, LXRα was significantly (*p* < 0.05) decreased at the level of both mRNA (Figure 5A) and protein (Figure 5E) in the liver of AFB1-challenged broiler chickens. BA supplementation protected chickens from AFB1-induced hepatic LXRα downregulation at both mRNA and protein levels. No significant alterations were determined in the hepatic mRNA expression of FXR (Figure 5B), RARα (Figure 5C), or RXRα (Figure 5D). Hepatic FXR protein content (Figure 5F) was significantly (*p* < 0.05) decreased in the AFB1 group, which was not reversed by BA supplementation. These results suggest that LXRα may be involved in BA-induced upregulation of detoxification genes in the liver of broiler chickens.

## 3. Discussion

In this study, we provide evidence that dietary supplementation with a compound BA preparation protected broiler chickens from AFB1-induced hepatotoxicity, including inflammation and oxidative stress. Quantitative analyses of AFB1 and its toxic metabolite AFBO in serum, liver, bile, and cecum content implicate improved AFB1 metabolism and AFB1/AFBO excretion from liver to gall bladder and cecum in BA-supplemented chickens. Additionally, the detoxification effects of BA compounds are associated with the upregulation of hepatic xenobiotic genes, including cytochrome P450s, GSTs, and BSEP, with the possible participation of LXRα-mediated gene regulation.

Oxidative stress and inflammation are common indicators of AFB1 toxicity [5,24]. Chickens show heightened sensitivity to AFB1, as evidenced by a significant increase in MDA, TNFα, and IFNγ levels in serum or spleen with 74 μg/kg AFB1 supplementation [25]. A similar study demonstrated a significant rise in duodenal MDA levels and a concurrent reduction in CAT and SOD content when broiler feed was supplemented with 5 mg/kg AFB1 [26]. In this study, the gavage of 250 μg/kg BW AFB1 to broilers triggered an inflammatory response and oxidative stress, marked by a reduction in hepatic CAT and SOD levels, coupled with an increase in the levels of TNFα, IL-1β, IFNγ, and iNOS in serum or liver. These findings are similar to previous studies [27]. Importantly, BAs exhibit significant efficacy in alleviating hepatic oxidative stress and inflammation across various animal models. For example, dietary supplementation with 80 mg/kg BA has been demonstrated to alleviate hepatic oxidative stress in broiler chickens under heat-stress conditions [28]. Similar effects have also been confirmed with T-2 toxin, where dietary supplementation with cholic acid contributes to mitigating oxidative stress and hepatic inflammatory cell infiltration in chickens [21]. In our study, we provide compelling evidence that dietary supplementation with BA protects broiler chickens from hepatic immune cell infiltration, IL-1β production, and oxidative stress induced by sub-chronic exposure to AFB1, highlighting the potential detoxification efficacy of BA in a scenario that closely resembles real conditions in broiler chicken farms.

Diverse mechanisms are reported underlying the mitigation effects of BA on hepatic oxidative stress and inflammation [16,29]. For example, BA supplementation significantly enhanced CAT and SOD activities in the liver while reducing MDA levels, thereby alleviating oxidative stress in largemouth bass exposed to a high-starch diet [30]. In vitro, pretreatment of rat hepatocytes with UDCA enhanced the levels of glutathione, protein thiol, and gamma-glutamyl-cysteine synthetase, alleviating H_2_O_2_-induced oxidative stress [31]. A recent comparative trial suggests that sheep BAs outperform pig BAs in promoting growth under heat stress, while pig BAs exhibit greater effectiveness in anti-oxidative stress and anti-inflammatory effects by elevating the levels of SOD and decreasing the contents of MDA, IL-1β, and TNFα [32]. In this experiment, BA exhibited effectiveness in alleviating AFB1-induced oxidative stress and inflammation in broiler chicken livers, correlating with a reduction in serum and hepatic AFB1 concentrations. Further analysis of AFB1 and AFBO in bile and cecum content revealed that BAs facilitate the metabolism and clearance of AFB1 through the hepatobiliary pathway. Similar findings align with previous research on different animal models and toxins. For instance, intravenous administration of UDCA in rats promoted the clearance of endotoxins through the hepatobiliary pathway [33]. Also, dietary supplementation of BA reduced AFB1 concentrations in the hepatopancreas of white shrimp [34]. Furthermore, supplementing the diet with 1% cholic acid improved T-2 toxin metabolism in chickens, leading to decreased T-2 levels in plasma and liver and mitigating T-2 toxin-induced liver inflammation and oxidative stress [21].

The turnover and clearance of xenobiotics are regulated by a series of bioactivation and detoxification enzymes as well as transporters responsible for hepatobiliary excretion [11,35]. In this study, broilers exposed to AFB1 exhibited disruptions in the expression of hepatic detoxification enzymes, including a decrease in phase I (CYP1A1, CYP1A2, and CYP3A4), phase II (GST), and phase III (BSEP). This observation is supported by previous literature documenting the down-regulation of CYP1A1 [23] and GST expression [34] due to AFB1 exposure. Notably, there was a significant increase in the expression of hepatic CYP1A1 and CYP2A6 in response to AFB1 [36,37]. As highlighted in the literature, the activity of AFB1 metabolic enzymes is influenced by various factors, including different animal species, age, and organs [11,38]. Additionally, the efflux of AFB1 and its metabolites primarily depends on the hepatobiliary pathway [39]. Our study further revealed a significant decrease in BSEP expression induced by AFB1. Moreover, we found that BA supplementation alleviated the AFB1-induced downregulation of genes involved in AFB1 bioactivation (CYP1A1 and CYP1A2), detoxification (GSTA2 and GSTCD), and excretion (BSEP) in the liver. This finding is in line with previous studies. For example, supplementation of 250 mg/kg BA compound significantly increased GST activity in AFB1-exposed white shrimps [34], and the addition of 1% cholic acid to the diet increased expression of P450 isoenzymes and BSEP in broiler chickens challenged with T-2 toxin [21]. For a deeper exploration of transcription factors in xenobiotic metabolism, we examined the expression levels of FXR, RARα, RXRα, and LXRα [40]. AFB1 notably downregulated LXRα gene expression and both LXRα and FXR protein levels in broiler livers in our study. Previous research has suggested a positive correlation between the expression of hepatic detoxification enzymes and FXR/LXRα levels [41]. It is noteworthy that the addition of BA to the diet alleviated the AFB1-induced decrease in LXRα expression. This implies that LXRα may play a role in regulating the expression of enzymes involved in AFB1 biotransformation and elimination in broiler chickens. Notably, the BA compound used in our study contains 73.2% hyodeoxycholic acid, recognized as an agonist for LXRα [42]. Additionally, the precise role of LXRα in the transcriptional regulation of xenobiotic metabolic genes in the chicken liver, especially in the face of the AFB1 challenge, requires further validation.

## 4. Conclusions

In conclusion, we provide evidence that a BA compound derived from porcine bile can effectively protect broiler chickens from hepatic inflammation induced by the sub-chronic AFB1 challenge. This is accomplished by transcriptional activation of genes involved in AFB1 biotransformation, detoxification, and hepatobiliary efflux, thereby promoting the turnover and clearance of AFB1 and its metabolite AFBO. Our findings suggest that the BA compound can be used as a reliable and effective strategy for reducing aflatoxicosis in chickens and possible residues of AFB1 and AFBO in chicken products.

## 5. Materials and Methods

### 5.1. Preparation of Bile Acids

The compound BA preparation used in the present study was produced by Shandong Longchang Animal Health Care Co., Ltd. (Dezhou, China) from porcine bile through a series of physicochemical methods, including saponification, decolorization, acidification, purification, and drying. The purity of this BA compound is 96.9%, comprising 73.2% hyodeoxycholic acid (HDCA), 19.8% chenodeoxycholic acid (CDCA), and 3.9% hyocholic acid (HCA).

### 5.2. Birds, Diets, and Experimental Design

One-day-old male Ross 308 broiler chickens (45 chicks) were reared in the experimental poultry house at Nanjing Agricultural University under standard conditions. After 4 days of adaptation, the chickens were randomly divided into three treatment groups of three replicates in each group (n = 15 birds per treatment). The CON and AFB1 groups were fed a basal diet (Table 1), and the AFB1 + BA group was fed a basal diet supplemented with 250 mg BA/kg feed (Figure 6). After 3 days of pre-feeding, (1) the chickens in the CON group only received solvent used in the gavage for the AFB1 treatment and diet without BA; (2) the chickens in the AFB1 group daily received AFB1 only (250 μg/kg BW) by gavage and diet without BA; and (3) the chickens in the AFB1 + BA groups were daily challenged with AFB1 (250 μg/kg BW) by gavage and fed a diet containing BA (250 mg BA/kg feed). The AFB1 solution for gavage administration was freshly prepared daily by being suspended in a saline:ethanol (95:5) mixture [43]. The doses of BA [44] and AFB1 [45], along with the intervention’s duration [23], were in accordance with the previous literature. Aflatoxin B1 (#MSS1003, with a purity of 99.4 ± 0.6%) was purchased from Qingdao Pribolab Pte., Ltd. (Qingdao, China). Water and feed were provided ad libitum. The light regime was 23 L:1 D, and the room temperature started at 35 °C for the first 3 days and gradually reduced by 3 °C per week until 23 °C, with the humidity kept at 50–60% throughout the experiment. On day 25, all chickens were sacrificed by rapid decapitation. Samples, including blood, liver, gallbladder, bile, and cecum content, were collected and stored for further analysis.

### 5.3. Histological Evaluation of Liver

Five chickens were randomly selected from each group (15 birds per group) for histological examination. The liver samples were fixed overnight in 4% paraformaldehyde and then embedded in paraffin. Five-micrometer slices were stained with hematoxylin and eosin (H&E). Inflammatory infiltrating cells surrounding the portal vein were manually counted under an optical microscope (Olympus-BX53, Olympus Corporation, Tokyo, Japan) and quantified following a previously described method [46].

### 5.4. Determination of Enzymes/Factors Related to Inflammation, Oxidative Stress and Detoxification

The activities of catalase (CAT, #BC0205), superoxide dismutase (SOD, #BC0175), and glutathione S-transferase (GST, #BC0350), as well as the total antioxidant capacity (T-AOC, #BC1315) and malondialdehyde (MDA, #BC0025) concentrations, were determined in liver lysates using respective kits from Beijing Solarbio Science & Technology Co., Ltd. (Beijing, China). The serum and hepatic concentrations of interleukin 1β (IL-1β, #MM-36910O1) were determined by using a commercial ELISA kit from Jiangsu Meimian Industrial Co., Ltd. (Nanjing, China) following the manufacturer’s instructions.

### 5.5. Determination of Total Bile Acids Concentration

Liver and cecal content were extracted using the method described previously [47]. The resulting extracts, together with the serum and bile samples, were used for determining the total bile acids (TBA) (#H101T) and cholesterol (CHOL) (#H202) concentrations with an automatic biochemical analyzer (Hitachi 7020, HITACHI, Tokyo, Japan) by using respective commercial kits purchased from Meinkang Biotech Co., Ltd. (Ningbo, China).

### 5.6. Determination of AFB1 and AFBO Concentrations

Liver and cecal content were extracted using the method following the manufacturer’s protocols as follows: frozen liver and cecal content were smashed and then added to a 70% methanol solution. The mixture was vigorously oscillated for 3 min and then filtered. The resulting extracts, together with the serum and bile samples, were used for determining the contents of AFB1 (#MM-1911O1) and AFBO (#MM-95274O1) with a full-wavelength microplate reader (Synergy 2, BioTek, Winooski, Vermont, USA) by using respective commercial ELISA kits purchased from Jiangsu Meimian Industrial Co., Ltd. (Shanghai, China).

### 5.7. Quantification of mRNA by Real-Time RT-PCR

Total RNA was extracted from the liver by using TRIzol reagent obtained from Tsingke Biotech Co., Ltd. (#TSP401, Nanjing, China). A total of 1 μg of total RNA was reverse transcribed into cDNA by using TransScript Uni All-in-One First-Strand cDNA Synthesis SuperMix obtained from TransGen Biotech Co., Ltd. (#AU341-02-V2, Nanjing, China). A total of 1 μL of diluted cDNA (1:20, *v*:*v*) was used for real-time PCR with PerfectStart Green qPCR SuperMix obtained from TransGen Biotech Co., Ltd. (#AQ601-02, Nanjing, China) using a Quant Studio™ 6 Flex Real-Time PCR System (Applied Biosystems, Wilmington, Massachusetts, USA). All primers (Table 2) were synthesized by Tsingke Biotech Co., Ltd. (Nanjing, China). The relative mRNA abundance was calculated with the 2^−ΔΔCT^ method using *PPIA* as an internal reference (Table 2 and Appendix A, and Appendix A).

### 5.8. Total Protein Extraction and Western Blotting

Protein was extracted from a 30 mg frozen liver sample as previously described [48]. The protein concentration was determined using the BCA protein assay kit obtained from Tsingke Biotech Co., Ltd. (#DQ111-01, Nanjing, China). Protein samples were loaded on 10% SDS-PAGE gels for electrophoresis. The primary antibodies used in Western blot analysis for BSEP (#A8467, ABclonal, Wuhan, China, diluted 1:1000), LXRα (#A2141, ABclonal, Wuhan, China, diluted 1:1000), FXR (#A8320, Wuhan, China, diluted 1:1000), and GAPDH (#AC001, Wuhan, China, diluted 1:100,000) were purchased from ABclonal Technology Co., Ltd. (Wuhan, China). β-actin, or GAPDH, was used as the internal control.

### 5.9. Statistics

All the data are presented as means ± SEM. A one-way ANOVA in IBM SPSS 20.0 software (SPSS Inc., Chicago, IL, USA) was used to test the significance of the differences among groups, followed by the least squares difference (LSD) for post hoc pairwise comparison. The differences were considered statistically significant when * *p* < 0.05, ** *p* < 0.01, and *** *p* < 0.001.

## Figures and Tables

**Figure 1 toxins-15-00694-f001:**
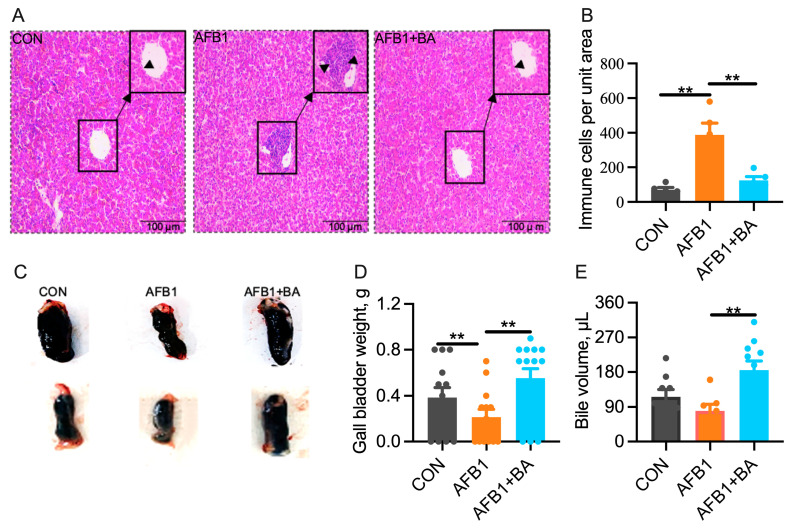
Effect of bile acids on the morphological changes in the liver and gallbladder in broiler chickens. (**A**) Hematoxylin and eosin (H&E) staining (scale bars = 100 μm and 400 μm). (**B**) Hepatic inflammatory cells per unit area. (**C**) Phenotypic presentation of the gall bladder. (**D**) Gall bladder weight. (**E**) Bile volume. Black arrows represent the infiltration of immune cells in the liver. Black rectangular borders indicate the zone of inflammatory cell infiltration. The differences were considered statistically significant when ** *p* < 0.01.

**Figure 2 toxins-15-00694-f002:**
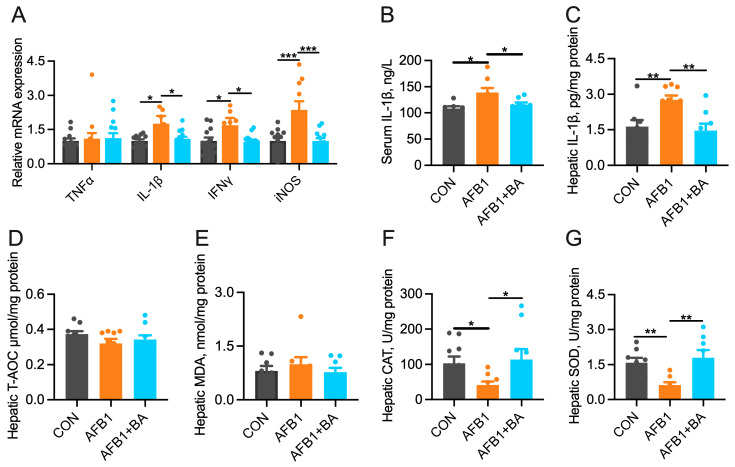
Effect of BA on the indicators of hepatic inflammation and oxidative stress in broiler chickens. (**A**) Hepatic mRNA expression of TNFα, IL-1β, IFNγ, and iNOS. (**B**) Serum IL-1β concentration. (**C**) Hepatic IL-1β concentration. (**D**) Hepatic T-AOC concentration. (**E**) Hepatic MDA concentration. (**F**) Hepatic CAT concentration. (**G**) Hepatic SOD concentration. The differences were considered statistically significant when * *p* < 0.05, ** *p* < 0.01, *** *p* < 0.001.

**Figure 3 toxins-15-00694-f003:**
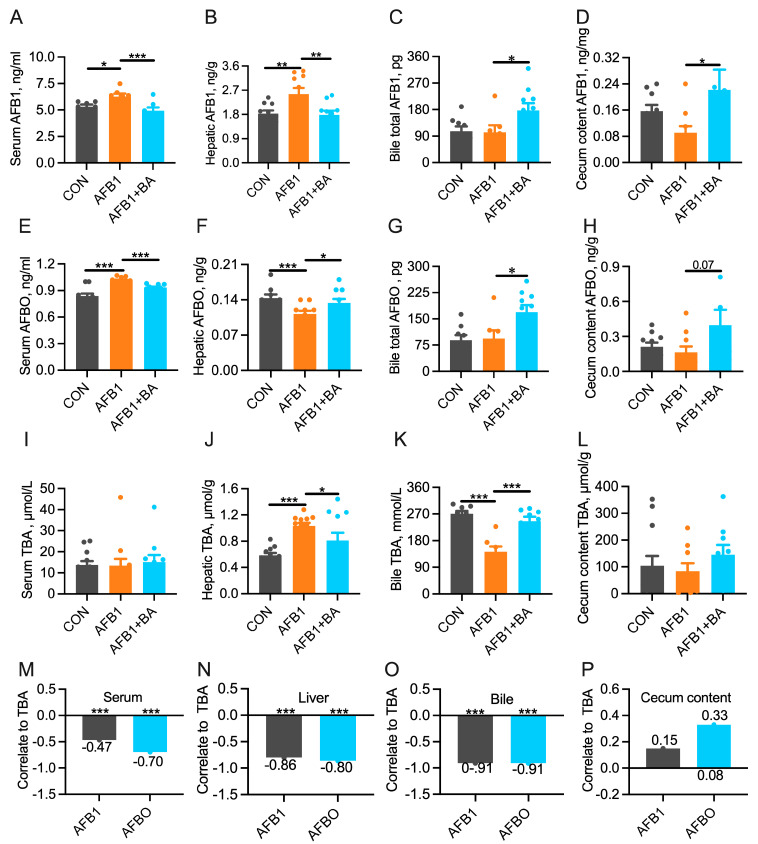
Effect of BA on the concentrations of AFB1, AFBO, and TBA in broiler chickens. (**A**) Serum AFB1 concentration. (**B**) Hepatic AFB1 concentration. (**C**) Bile total AFB1 content. (**D**) Cecum content AFB1 concentration. (**E**) Serum AFBO concentration. (**F**) Hepatic AFBO concentration. (**G**) Bile total AFBO content. (**H**) Cecum content AFBO concentration. (**I**) Serum TBA concentration. (**J**) Hepatic TBA concentration. (**K**) Bile TBA concentration. (**L**) Cecum content TBA concentration. (**M**) Regression analyses between AFB1/AFBO and TBA concentrations in serum. (**N**) Regression analyses between AFB1/AFBO and TBA concentrations in the liver. (**O**) Regression analyses between AFB1/AFBO and TBA concentrations in bile. (**P**) Regression analyses between AFB1/AFBO and TBA concentrations in cecum content. The differences were considered statistically significant when * *p* < 0.05, ** *p* < 0.01, *** *p* < 0.001.

**Figure 4 toxins-15-00694-f004:**
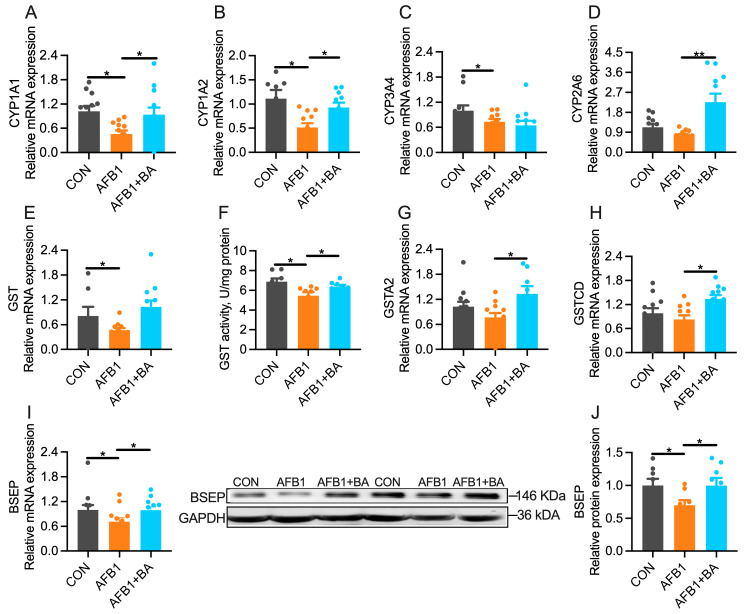
Effect of BA on AFB1-induced hepatic levels of hepatic phase I, II enzymes, and transporters in broiler chickens. (**A**) Hepatic mRNA expression of CYP1A1. (**B**) Hepatic mRNA expression of CYP1A2. (**C**) Hepatic mRNA expression of CYP3A4. (**D**) Hepatic mRNA expression of CYP2A6. (**E**) Hepatic mRNA expression of GST. (**F**) Hepatic GST activity. (**G**) Hepatic mRNA expression of GSTA2. (**H**) Hepatic mRNA expression of GSTCD. (**I**) Hepatic mRNA expression of BSEP. (**J**) Hepatic protein content of BESP. The differences were considered statistically significant when * *p* < 0.05, ** *p* < 0.01.

**Figure 5 toxins-15-00694-f005:**
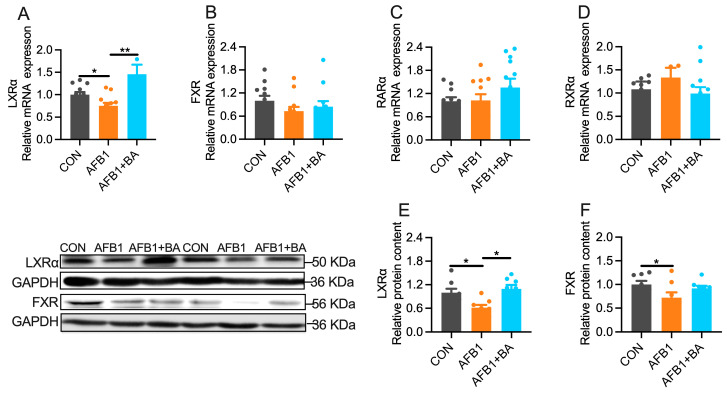
Effect of BA on AFB1-induced hepatic nuclear receptors in broiler chickens. (**A**) Hepatic mRNA expression of LXRα. (**B**) Hepatic mRNA expression of FXR. (**C**) Hepatic mRNA expression of RARα. (**D**) Hepatic mRNA expression of RXRα. (**E**) Hepatic protein content of LXRα. (**F**) Hepatic protein content of FXR. The differences were considered statistically significant when * *p* < 0.05, ** *p* < 0.01.

**Figure 6 toxins-15-00694-f006:**
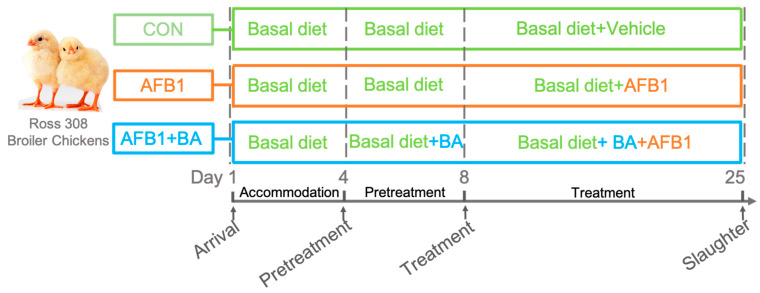
Flow chart of the experimental design.

**Table 1 toxins-15-00694-t001:** Ingredients and nutrient composition of the diets for broiler chickens.

Items	CON	AFB1	AFB1 + BA
Ingredients, %			
Wheat	25	25	25
Corn powder	20	20	20
Corn meal	14	14	14
Soybean meal	12	12	12
Rice powder	10	10	10
Corn gluten powder	4	4	4
Feather powder	2	2	2
DDGS	2	2	2
Candy meal	1	1	1
Rapeseed meal	3	3	3
Corn	2	2	2
Soybean oil	0.5	0.5	0.5
Limestone	1.2	1.2	1.2
Calcium hydrogen phosphate	1.4	1.4	1.4
Sodium chloride	0.29	0.29	0.29
Lysine	0.81	0.81	0.81
Methionine	0.13	0.13	0.13
Tryptophan	0.01	0.01	0.01
Choline	0.1	0.1	0.1
Premix	0.56	0.56	0.56
Bile acids, mg/kg	0	0	250
Calculated nutrient composition, %			
Metabolizable energy, kcal/kg	3100	3100	3100
Crude protein	21	21	21
Calcium	1.30	1.30	1.30
Available phospholipid	0.80	0.80	0.80
Sodium chloride	0.80	0.80	0.80
Methionine	0.90	0.90	0.90
Lysine	0.60	0.60	0.60

Note: The premix was composed of the following per kg diet: Vitamin A 9000 IU; Vitamin D3, 3600 IU; Vitamin E, 12 IU; Vitamin K_3_, 3.00 mg; Vitamin B1, 2.00 mg; Vitamin B2, 6.90 mg; Vitamin B6, 2.70 mg; Vitamin B12, 0.02 mg; D-Biotin, 0.23 mg; Nicotinic acid, 31.20 mg; Folic acid, 1.00 mg; Vitamin B5, 10.80 mg; Choline chloride, 0.30 g; Fe (as FeSO_4_), 60.00 mg; Cu (as CuSO_4_), 12.00 mg; Mn, 90.00 mg; Zn (as ZnSO_4_), 90.00 mg; I (as KI), 0.80 mg; Se (as Na_2_SeO_3_), 0.30 mg.

**Table 2 toxins-15-00694-t002:** Nucleotide sequences of specific primers.

Target Genes	GenBank Accession	Primer Sequences (5′ to 3′)	PCR Product (bp)
*CYP1A1*	NM_205147.2	F: TCGTCAACGACCTCTTTGGG R: CAAGGCAGCGTACATCATGC	77
*CYP1A2*	NM_205146.3	F: GAGGATCCTGTGTGGTTCCG R: AACTAAGGGGAAGCGTGGTG	132
*CYP3A4*	NM_001329508.2	F: CCCTGCAAAGACACTCCGAT R: CTGGGGCCAAGGAATTGTCA	299
*CYP2A6*	PMID: 28377720	F: CTGCAGAGAATGGCATGAAG R: CCTGCAAGACTGCAAGGAA	110
*TNFα*	NM_204267.2	F: CTGAGGCATTTGGAAGCAGC R: GACAGGGTAGGGGTGAGGAT	208
*IL-1β*	NM_000576.3	F: AACCTCTTCGAGGCACAAGG R: AGCCATCATTTCACTGGCGA	122
*INF-γ*	NM_205149.2	F: CACTGACAAGTCAAAGCCGC R: GCATCTCCTCTGAGACTGGC	232
*iNOS*	NM_204961.2	F: CCAGCTGATTGGGTGTGGAT R: CCTACGGGTCTCATCATGCC	150
*GST*	NM_001001777.2	F: AGGAAACCACGCCTAGAGGA R: TTTCATCCAGTGTACCGCCT	213
*GSTA2*	NM_001001776.2	F: GGCGCTGCAGTCAAGCTC R: TCCTCGAATTCAACCCCAGC	282
*GSTCD*	XM_015276567.4	F: TCTTGATCGAGCATGGGCTG R: TAAACTGGGGTGCCCACAAA	111
*BSEP*	XM_052674968.1	F: ACCCAGGTGCAAATAGCCAAT R: GCACCCAAACACTTCCCATC	77
*FXR*	NM_001396910.1	F: AAAAGCCTAGACTGGGCCAC R: GCCAACATGCCCATTTGCTT	253
*LXRα*	NM_204542.3	F: CTTCCGCTGGTACTTCCTTTC R: CAGGAGGCCTTCTTCAAGGT	70
*RXRα*	XM_003642291.5	F: CCCTCTGAACACCAGGTGAC R: ACAGAATCCAGCAGGAGCAC	280
*RARα*	NM_204536.1	F: TCACCCCCTACGCCTTTTTC R: AGGATTTGTCCTGGCAGACG	225
*PPIA*	NM_001166326.2	F: GTGACTTTACGCGCCACAAC R: TTGCTCGTCTTGCCGTCTTT	268

## Data Availability

The data presented in this study are available within this article.

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
