# Peer review of "Bile Acids Promote Hepatic Biotransformation and Excretion of Aflatoxin B1 in Broiler Chickens"

_toxins, 2023, doi:10.3390/toxins15120694_

Round 1

Reviewer 1 Report

Comments and Suggestions for Authors

·       Table 1 does not contain details of AFB1 amount for AFB1 group, if AFB1 was not added then how it was induced?

·       Table1 also do not have details of BA, all items and ingredients are same then what is the major difference between these 3 groups? revise Table provide accurate detail

·       BA was added into the feed after induction of AFB1 or not

·       For histological exam author selected 5 broilers from each group for samples. However, total 5 birds was used in each group, it seems there was no any mortality in the groups throughout the experiment?

·       Why author used PPIA as a reference gene? Reason?

·       Table 2 CYP2A6 gene BP detail is missing.

·       author used β-actin or GAPDH in WB as a reference gene it is different from qPCR analysis ? Why vice versa?  

·       Figure 1 caption is difficult to understand. Why detail of method is again added, number of samples is provided in the caption? Which is already provided, what findings author found  in the figure? Provide them only, what is the meaning of arrow symbol?

·       How infiltrating immune cells was calculated? Explain.

·       Figure 1 is very large size, separate results of histology and oxidative stress parameters and explain them separately

·       Line 84 - 88 antioxidant results are poorly written and difficult to understand. Explain what you found in control, AFB1 first then AFB1+BA groups, explain in the proper sequence according to groups, it is mixture of liver and serum, ?

·       Results of Dietary BA supplementation promote the turnover of AFB1/AFBO and bile acid are difficult to understand as well and very confusing. Serum, bile and cecum parameters should be separated for each content, if possible make them in brief with two or three figures

·       The author only found BSEP protein levels but not other gene? Why only 1 protein is selected not others?

·       Blot figure is cropped and not clear, provide original figure

·       All figure legends need revisions, all details should be deleted which are already provided in material and method section, necessary to provide details of each section of figure a, b, c, etc. or different symbols, arrows meaning and scale bar in the figures. Other details need to omit.

·       Results Dietary BA supplementation enhances hepatic expression of LXRα explains that No significant alterations were determined in hepatic mRNA expression of FXR (Figure 4B), RARα (Figure 4C), or RXRα (Figure 4D) was found and FXR protein was not reversed by BA supplementation so how author justifies that BA treatment restores detoxification genes?

·       The results are not supported by the overall discussion of the manuscript, and it requires extensive revisions based on each parameter.

·       Revise the paper and highlight all changes in red font to view the author's changes.

Comments on the Quality of English Language

Moderate editing of English language required

Author Response

Reviewer 1

  1. Table 1 does not contain details of AFB1 amount for AFB1 group, if AFB1 was not added then how it was induced?

Reply: Thanks for the comments. AFB1 was administered via gavage and is not included in Table 1, as mentioned on page 9, line 287-289.

  1. Table 1 also do not have details of BA; all items and ingredients are same then what is the major difference between these 3 groups? revise Table provide accurate detail

Reply: Thanks for the comments. We detailed the BA supplementation in Table 1 of the revised manuscript, specified on page 10, line 298.

  1. BA was added into the feed after induction of AFB1 or not

Reply: Thanks for the comments. BA was added to the feed three days prior to the induction of AFB1, as shown on page 9, line 281-283

  1. For histological exam author selected 5 broilers from each group for samples. However, total 5 birds were used in each group, it seems there was no any mortality in the groups throughout the experiment?

Reply: Thanks for the comments. Firstly, no chicken died in any of the groups throughout the experiment. Secondly, we realize that this sentence is not clear and revised it in the revised manuscript on page 11, lines 308-309, as follows:

“Five chickens were randomly selected from each group (15 birds per group) for histological examination.”

  1. Why author used PPIA as a reference gene? Reason?

Reply: Thanks for your comments. We have selected five frequently used candidates and assessed their stability using Normfinder (https://seqyuan.shinyapps.io/seqyuan_prosper/). Ultimately, in our present study, PPIA demonstrated the highest level of stability among the five housekeeping genes. The raw data and analysis results for these housekeeping genes are provided in Table S1 and Figure S3 of the supplementary material, as indicated on page 12, line 346.

  1. Table 2 CYP2A6 gene BP detail is missing.

Reply: Thanks for your comments. We have provided the “BP” detail of the CYP2A6 in Table 2 of the revised manuscript on page 12, line 347.

  1. author used β-actin or GAPDH in WB as a reference gene it is different from qPCR analysis? Why vice versa?  

Reply: Thanks for your comments. GAPDH, a frequently utilized antibody, demonstrated consistent performance in the present experiment. However, due to the unavailability of a PPIA antibody, a direct comparative experiment between GAPDH and PPIA antibodies was not undertaken. In qPCR analysis, PPIA, chosen for its stability through Normfinder (https://seqyuan.shinyapps.io/seqyuan_prosper/) under our experimental conditions, ensures the reliability of the housekeeping gene. This decision underscores the customized selection of suitable reference genes for each analytical method.

  1. Figure 1 caption is difficult to understand. Why detail of method is again added, number of samples is provided in the caption? Which is already provided, what findings author found in the figure? Provide them only, what is the meaning of arrow symbol?

Reply: Thanks for the comments. We have revised the caption of Figure 1 in accordance with your suggestions, as mentioned on page 3, lines 81-85. Additionally, we have supplied detailed descriptions of the arrow symbols and black rectangular borders on page 3, lines 85-86, as follows:

“Black arrows represent the infiltration of immune cells in liver. Black rectangular borders indicate the zone of the inflammatory cell infiltration.”

  1. How infiltrating immune cells was calculated? Explain.

Reply: Thanks for the comments. The quantification of infiltrating immune cells was conducted using the method described previously (Gadd et al., 2014). Briefly, for each section from the broiler liver, we took five hepatic sinusoids for further analysis. With the hepatic sinusoid as the focal point, we captured five microscopic fields around each lobule. Subsequently, manual cell counts were performed for immune cells in each captured field. The total count of immune cells across the five fields was then averaged, yielding the average number of immune cells for each hepatic sinusoid. This systematic approach ensured a comprehensive assessment of immune cell distribution within each sample, contributing to a more accurate understanding of the immune response in the hepatic sinusoid.

Gadd, V.L.; Skoien, R.; Powell, E.E.; Fagan, K.J; Winterford, C.; Horsfall, L.; Irvine, K.; and Clouston, A.D. The portal inflammatory infiltrate and ductular reaction in human nonalcoholic fatty liver disease. Hepatology. 2014, 59, 1393-405. doi: 10.1002/hep.26937.

  1. Figure 1 is very large size, separate results of histology and oxidative stress parameters and explain them separately.

Reply: Thanks for the comments. We have separated "Figure 1" into two distinct figures, now labeled as Figure 1 and Figure 2, as shown on page 3, line 81-86 and page 4, line 99-103. Additionally, we have also made corresponding modifications to the figure numbering throughout the entire manuscript in the revised manuscript.

  1. Line 84-88 antioxidant results are poorly written and difficult to understand. Explain what you found in control, AFB1 first then AFB1+BA groups, explain in the proper sequence according to groups, it is mixture of liver and serum?

Reply: Thanks for the comments. Firstly, we confirmed the assessment of the activity (or content) of the oxidative stress markers in the liver, including T-AOC, MDA, CAT, and SOD. Secondly, we have revised the sentences of the “antioxidant results” in the manuscript on page 3, lines 92-97, as follows:

“Additionally, compared to the CON group, there were no significant differences in T-AOC (Figure 2D) and MDA (Figure 2E), while AFB1 induced a significant reduction (P < 0.05) in the activities of CAT (Figure 2F) and SOD (Figure 2G). Furthermore, the AFB1+BA group exhibited a significant increase (P < 0.05) in CAT and SOD activities, with no significant differences in T-AOC and MDA when compared to the AFB1 group.”

  1. Results of Dietary BA supplementation promote the turnover of AFB1/AFBO and bile acid are difficult to understand as well and very confusing. Serum, bile and cecum parameters should be separated for each content, if possible, make them in brief with two or three figures

Reply: Thanks for the comments. While we understand your suggestion to present serum, bile, and cecum parameters separately, we currently may not be inclined to split the comprehensive figure for the sake of overall coherence. However, to enhance result clarity, we will carefully review the labeling and descriptions of the figure to ensure readers find it easier to comprehend the information presented. We will ensure to provide ample explanations in the text to clarify the intricacies of the result presentation, as shown in the revised manuscript on page 4, line 105-119, as follow:

“Compared to the CON group, the AFB1 group exhibited a significant increase (P < 0.05) in AFB1 levels in serum (Figure 3A) and liver (Figure 3B). In contrast, BA supplementation significantly reduced (P < 0.05) AFB1 levels in both serum and liver compared to the AFB1 group. Additionally, in comparison to the AFB1 group, the AFB1+BA group showed a significant increase (P < 0.05) in AFB1 concentrations in bile (Figure 3C) and cecum content (Figure 3D), suggesting that BA supplementation pro-motes the excretion of AFB1. Interestingly, AFBO displayed a distinct metabolic pattern. Compared to the CON group, AFBO concentrations significantly increased (P < 0.05) in serum (Figure 3E) but decreased significantly (P < 0.05) in the liver (Figure 3F) in the AFB1 group. On the contrary, compared with the AFB1 group, the AFB1+BA group exhibited a significant decrease (P < 0.05) in serum and increase (P < 0.05) in the liver in AFBO concentration. Furthermore, compared to the AFB1 group, the AFB1+BA group showed a significant increase (P < 0.05) in AFBO concentrations in both bile (Figure 3G) and cecum content (Figure 3H) (P < 0.05). These results suggest that dietary BA supplementation modulates the metabolism and excretion of AFB1 in the liver.”

  1. The author only found BSEP protein levels but no other gene? Why only 1 protein is selected not others?      

Reply: Thanks for the comments. We realize the significance of data f pertaining to key hepatic efflux transporters (MRP2/3/4 and MDR1) to enhance the robustness of the findings in this study. These results are now included in Figure S2 in the revised manuscript on page 6, lines 155-157.  

  1. Blot figure is cropped and not clear, provide original figure

Reply: Thanks for the comments. We have supplied the uploaded supplementary materials titled “Supplemental materials”.                                                                                                            

  1. All figure legends need revisions, all details should be deleted which are already provided in material and method section, necessary to provide details of each section of figure a, b, c, etc. or different symbols, arrows meaning and scale bar in the figures. Other details need to omit.

Reply: Thanks for the comments. We have made modifications to the revised manuscript on page 3, line 81-86, page 4, line 99-103, page 5, line 132-136, page 6, line 162-164, and page 7, line 165-167, line 180-183.

  1. Results Dietary BA supplementation enhances hepatic expression of LXRα explains that no significant alterations were determined in hepatic mRNA expression of FXR (Figure 4B), RARα (Figure 4C), or RXRα (Figure 4D) was found and FXR protein was not reversed by BA supplementation so how author justifies that BA treatment restores detoxification genes?

Reply: Thanks for the comments. In our study, the introduction of dietary bile acid (BA) supplementation led to a noteworthy increase in both LXRα gene and protein levels, corresponding to observed changes in phase I, II, and III enzymes. Existing literature supports the pivotal role of LXRα in governing the expression of hepatic phase I (Araki et al., 2012), II (Gong et al., 2009), and III (Uppal et al., 2007) detoxifying enzymes. Notably, an upward trend in FXR protein levels was observed with BA supplementation compared to the AFB1 group. This suggests a complex interplay between various transcription factors and signaling pathways, potentially influencing the overall detoxification process. Our interpretation of these findings leads us to propose that LXRα may play a role in regulating AFB1 detoxification in broilers.

(1) Araki, K.; Watanabe, K.; Yamazoe, Y.; Yoshinari. Liver X receptor α bidirectionally transactivates human CYP1A1 and CYP1A2 through two cis-elements common to both genes. Toxicol Lett. 2012, 215, 16-24. doi: 10.1016/j.toxlet.2012.09.021.

(2) Gong, H.; He, J.; Lee, J.H.; Mallick, E.; Gao, X.; Li, S.; Homanics, GE.; Xie, W. Activation of the liver X receptor prevents lipopolysaccharide-induced lung injury. J Biol Chem. 2009, 284, 30113-30121. doi: 10.1074/jbc.M109.047753.

(2) Uppal, H.; Saini, S.P.; Moschetta, A.; Mu, Y.; Zhou, J.; Gong, H.; Zhai, Y.; Ren, S. Michalopoulos GK, Mangelsdorf DJ, Xie W. Activation of LXRs prevents bile acid toxicity and cholestasis in female mice. Hepatology. 2007, 45, 422-432. doi: 10.1002/hep.21494.

  1. The results are not supported by the overall discussion of the manuscript, and it requires extensive revisions based on each parameter.

Reply: Thanks for the comments. We have made modifications to the revised manuscript on the page 7, line 194-195, page 8, line 196-207, line 213-223, line 234-243, and page 9, line 249-255.

Reviewer 2 Report

Comments and Suggestions for Authors

The article is interesting, aiming to investigate whether dietary supplementation of a compound bile acid preparation could alleviate AFB1-induced inflammation and oxidative stress and whether this is associated with LXRα-related regulation of hepatic detoxification genes, including bioactivation (phase I) and detoxification (phase II) enzymes, 65 as well as ABC transporters responsible for AFB1 excretion..

The topic is relevant updating research on AFB1 detoxification.

The methodology is very modern and complex, investigations being conducted from usual histology to the very actual molecular biology

The conclusions are adequate, the authors providing evidence that a BA compound derived from porcine bile can effectively protect broiler chickens from hepatic inflammation induced by sub-chronic AFB1 challenge..

I suggest some small corrections

1.       Line 28, references [3,4] are doubled

2.       Line 71 authors are mentioning (data not shown). Why? Dat should be provided at least as supplementary material

3.       Introduction could be improved, mentioning some other methods of AFB1 detoxification.

Eg., see Carmen Solcan, Mihaela Gogu, Viorel Floristean, Bogdan Oprisan, and Gheorghe Solcan, The hepatoprotective effect of sea buckthorn (Hippophae rhamnoides) berries on induced aflatoxin B1 poisoning in chickens, Poultry Science, 2013; 92(4):966-974, doi: 10.3382/ps.2012-02572

4.       The references are appropriate but when are edited the authors should follow carefully the Instructions for authors for MDPI journals, including doi code.

.

Author Response

  1. Line 28, references [3,4] are doubled

Reply: Thanks for your comments. We have removed a citation '[3,4]' from page 1, line 28.

  1. Line 71 authors are mentioning (data not shown). Why? Data should be provided at least as supplementary material

Reply: Thanks for your comments. We realize that this data necessary to support the results in this study. We have provided the data as Figure S1 in the revised manuscript on page 2, line 74.

  1. Introduction could be improved, mentioning some other methods of AFB1 detoxification. Eg., see Carmen Solcan, Mihaela Gogu, Viorel Floristean, Bogdan Oprisan, and Gheorghe Solcan, the hepatoprotective effect of sea buckthorn (Hippophae rhamnoides) berries on induced aflatoxin B1 poisoning in chickens, Poultry Science, 2013; 92(4):966-974, doi: 10.3382/ps.2012-02572.

Reply: Thanks for your comments. We have included additional methods of AFB1 detoxification as requested in the revised manuscript on page 2, lines 58-61, as follows:

This study aimed to reducing aflatoxicosis in broilers, with reported evidence indicating that sea buckthorn oil [22] and hydrated sodium calcium aluminosilicate [23] through reducing the residues of AFB1, stimulating hepatic protein synthesis, or enhancing immune function in the liver.”

(1) Solcan. C.; Gogu, M.; Floristean, V.; Oprisan, B.; Solcan, G.; The hepatoprotective effect of sea buckthorn (Hippophae rhamnoides) berries on induced aflatoxin B1 poisoning in chickens. Poult Sci. 2013, 92, 966-974. doi: 10.3382/ps.2012-02572.

(2) Chen, X.; Horn, N.; Applegate, T.J. Efficiency of hydrated sodium calcium aluminosilicate to ameliorate the adverse effects of graded levels of aflatoxin B1 in broiler chicks. Poult Sci. 2014, 93, 2037-2047. doi: 10.3382/ps.2014-03984.

  1. The references are appropriate but when are edited the authors should follow carefully the Instructions for authors for MDPI journals, including doi code.

Reply: Thanks for your comments. We have revised according to your suggestions on page 13, line 368-401, page 14, line 402-458, and page 15, line 459-487.

Round 2

Reviewer 1 Report

Comments and Suggestions for Authors

accept